# LEARNING SINGLE-COMPONENT DISCRIMINATIVE REPRESENTATIONS VIA MAXIMAL CODING RATE REDUCTION

## ABSTRACT

The recently proposed maximal coding rate reduction principle (MCR$^2$) offers a promising theoretical framework for interpreting modern deep networks through the lens of data compression and discriminative representation. It maps high-dimensional multi-class data into mutually orthogonal linear subspaces, with each subspace capturing as many structural details of its class as possible. In this work, we show that such structural maximization not only increases model sensitivity to feature noise but also hinders generalization. In contrast, we argue that retaining only the single most discriminative structural component per class improves both generalization and robustness to feature noise, while preserving the desirable properties of MCR$^2$, such as robustness to label noise and resistance to catastrophic forgetting. We formalize this approach as a new framework termed SiMCoding and validate it extensively across supervised learning, white-box architectures, and incremental learning on diverse datasets. The superior performance of SiMCoding highlights its potential as a strong alternative for medium-scale classification tasks, particularly under label and feature noise.

## 1 INTRODUCTION

Numerous research efforts have sought to demystify the black-box nature of deep learning. Among these, an influential direction is the principle of *Maximal Coding Rate Reduction* (MCR$^2$) (Yu et al., 2020), which reformulates the learning objective to explicitly capture the low-dimensional structures underlying high-dimensional data, rather than focusing primarily on label fitting. MCR$^2$ is grounded in the manifold hypothesis, which posits that although data points $\boldsymbol{x} \in \mathbb{R}^D$ are observed in a high-dimensional ambient space, their variability is largely confined to a union of low-dimensional submanifolds, $\mathcal{M} = \bigcup_{i=1}^{K} \mathcal{M}_i$, as illustrated in Figure 1 (Hein & Audibert, 2005; Spigler et al., 2019; Pope et al., 2021; Wright & Ma, 2021). Each submanifold $\mathcal{M}_i$ corresponds to a semantic class or cluster, and the central objective of MCR$^2$ is to faithfully uncover and effectively organize these structures in the feature space.

As a foundational concept in information theory, the lossy *coding rate* $R(\boldsymbol{Z}, \epsilon)$ quantifies **the volume of a distribution** or its finite set $\boldsymbol{Z}$, up to a precision $\epsilon$ (Rissanen, 1998; Cover, 1999; Ma et al., 2007): a lower coding rate indicates a more compact set. What distinguishes the coding rate from other classical concepts in information theory, such as entropy and mutual information, is that it serves as a well-defined measure of distribution compactness even for degenerate distributions, which commonly arise in data with relatively low intrinsic dimensionality. Formally, in a $K$-class classification task, let the features of the $i$-th class be denoted by $\boldsymbol{Z}_i \in \mathbb{R}^{d \times m_i}$, and define the overall feature set as $\boldsymbol{Z} = \cup_{i=1}^{K} \boldsymbol{Z}_i$. The MCR$^2$ framework aims to **maximize the volume** of the overall feature set $\boldsymbol{Z}$ while simultaneously **minimizing the volumes** of the individual cluster sets $\boldsymbol{Z}_i$. This simple mechanism effectively maps high-dimensional data into a compact and structured low-dimensional representation, as depicted in Figure 1:

1. **Discriminative representation:** Features of each class $\boldsymbol{Z}_i$ are compressed into a low-dimensional linear subspace $\mathcal{S}_i$, and these subspaces are mutually orthogonal, i.e., $\boldsymbol{Z}_i \boldsymbol{Z}_j^\top = \boldsymbol{0}$ for all $i \neq j$.
2. **Diverse representation:** The dimensionality (or variance) of features of each class is maximized subject to the constraint of the representation space $\mathbb{R}^d$, i.e., $\sum_{i=1}^{K} \text{rank}(\boldsymbol{Z}_i) = d$.

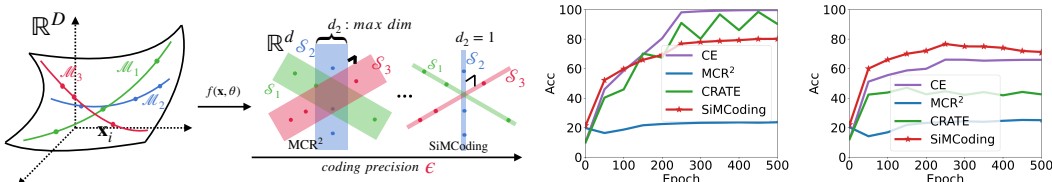

Figure 1: (Left) MCR$^2$ maps data $x_i$, typically distributed over nonlinear low-dimensional submanifolds $\mathcal{M}_i$, onto mutually orthogonal linear subspaces $\mathcal{S}_i$ with maximal dimensionality, whereas SiMCoding enforces each subspace to be one-dimensional. (Middle and Right) Training and test accuracy on CIFAR-20 with 20% randomly corrupted labels.

Note that, to achieve the second property, the features must be encoded with high precision $\epsilon$, enabling the model to capture as many structural details as possible and thereby allowing each class-specific feature set $\mathbf{Z}_i$ to attain its maximal dimensionality, i.e., maximal structural components.

Owing to its simplicity and conceptual interpretability, MCR$^2$ has emerged as an influential framework in representation learning and has been applied across diverse settings. It has inspired the design of interpretable white-box network architectures (Chan et al., 2022; Pai et al., 2023; Yu et al., 2024a; Yang et al., 2024) and efficient self-attention modules (Wu et al., 2024). It has also been explored in incremental learning (Wu et al., 2021; Tong et al., 2023), generative modelling (Dai et al., 2022), and unsupervised learning (Tong et al., 2022; Chu et al., 2024; Wu et al., 2025), among others.

However, in this work, we question whether it is truly necessary for a model to maximize structural details across different learning settings. In unsupervised learning, where labels are unavailable, it is natural for the model to retain as much structural information as possible in order to deeply uncover the underlying structure and subsequently cluster the samples. In contrast, in supervised learning, where labels provide guidance, preserving only a single discriminative structural component may be sufficient for accurate classification. As a thought experiment, consider classifying images of the digits $0$ and $1$. Recognizing their overall outlines is sufficient for the task, whereas fine-grained structural details—such as the precise curvature of a $0$ or the thickness of a $1$—are unnecessary. Even though in practice the features learned by neural networks are often highly abstract and difficult to interpret directly (Zeiler & Fergus, 2014; Chen et al., 2023), this example illustrates that in supervised classification tasks, high-level features may only need to preserve a single discriminative structural component rather than all structural details of the input.

The main contributions of this work are summarized as follows:

- While MCR$^2$ has achieved remarkable success, particularly in unsupervised learning, we find that in supervised classification its pursuit of maximal structural detail leads to severe underfitting and poor generalization. As shown in Figure 1, MCR$^2$ struggles to fit CIFAR-20 dataset (with 20% randomly corrupted labels) (Krizhevsky et al., 2009) effectively and exhibits weak generalization. Moreover, it is highly vulnerable to input noise (Table 2).

- To address these issues, we propose learning only the single most discriminative structural component for each class, rather than maximizing all structural detailsza. We term this approach **Si**ngle-component **M**aximal **Coding** rate reduction (SiMCoding).

  Specifically, we first provide a theoretical analysis showing how the coding precision $\epsilon$ determines the extent to which the model emphasizes structural details, formally corresponding to the varying dimensionality of each class subspace $\mathcal{S}_i$. This analysis further reveals that $\epsilon$ can be pre-specified to ensure that each class attains its minimal one dimensional subspace. As an important byproduct, this theory removes the need to tune $\epsilon$ as a hyperparameter, thereby significantly reducing the burden of applying SiMCoding.

- We validate SiMCoding across a wide range of datasets and learning settings. Experiments show that SiMCoding matches the fitting ability and generalization of the widely used cross-entropy framework, while exhibiting substantially stronger robustness to label noise. As shown in Figure 1, on CIFAR-20 with 20% randomly corrupted labels, the training accuracy of SiMCoding plateaus near 80%, indicating that it fits only the correctly labeled samples. Moreover, despite adopting the opposite strategy of retaining only a single structural com-

ponent per class, SiMCoding preserves key properties of MCR$^2$, including robustness to catastrophic forgetting in incremental learning setting.

- We analyse the computational complexity of SiMCoding and conclude that, despite potential limitations, it remains a strong alternative for classification on datasets with a moderate number of classes (e.g., $K \leqslant 100$), particularly in the presence of feature or label noise.

## 2 METHOD

**Representation learning and the MCR$^2$ principle.** We are given data $\boldsymbol{X} = [\boldsymbol{x}_1, \boldsymbol{x}_2, \ldots, \boldsymbol{x}_m] \in \mathbb{R}^{D \times m}$ consisting of $m$ samples from $K$ classes. The aim of deep representation learning is to transform high-dimensional data into low-dimensional features that capture intrinsic properties such as structure and geometry to facilitate downstream tasks such as classification. A widely used viewpoint, often referred to as the *manifold hypothesis*, suggests that each class lies on a low-dimensional submanifold $\mathcal{M}_i$, and that the entire dataset is concentrated near the union $\mathcal{M} = \cup_{i=1}^K \mathcal{M}_i$ (Hein & Audibert, 2005; Spigler et al., 2019; Pope et al., 2021; Wright & Ma, 2021). This motivates seeking features $\boldsymbol{z}_i \in \mathbb{R}^d$ with $d \ll D$ that retain this structure while discarding redundant variability. To obtain such features, one typically employs a nonlinear map $f_{\boldsymbol{\Theta}} : \mathbb{R}^D \to \mathbb{R}^d$ parametrized by neural network weights $\boldsymbol{\Theta}$:

$$\boldsymbol{x} \mapsto \boldsymbol{z} = f_{\boldsymbol{\Theta}}(\boldsymbol{x}),$$

and collects the feature matrix $\boldsymbol{Z} = [\boldsymbol{z}_1, \ldots, \boldsymbol{z}_m] \in \mathbb{R}^{d \times m}$. Desirable features should not only align with the underlying class structure but also admit a compact, structured and interpretable form.

The principle of *Maximal Coding Rate Reduction* (MCR$^2$) (Yu et al., 2020; Chan et al., 2022) provides an information-theoretic criterion for achieving this goal. It simultaneously maximizes the overall volume of all features to encourage separation across classes (*expansion*) and minimizes the average volume of each class to promote compactness (*compression*). Specifically, let $\boldsymbol{\Pi} = \{\boldsymbol{\Pi}_i \in \mathbb{R}^{m \times m}\}_{i=1}^K$ denote a set of diagonal matrices, where each diagonal entry $\boldsymbol{\Pi}_i(j,j)$ specifies the probability that sample $j$ belongs to class $i$. The MCR$^2$ framework then seeks to optimise

$$\max_{\boldsymbol{Z}, \boldsymbol{\Pi}} \ \Delta\mathcal{R}(\boldsymbol{Z}, \boldsymbol{\Pi}, \epsilon) = \underbrace{\frac{1}{2} \log\det\left(\boldsymbol{I} + \frac{d}{m\epsilon^2} \boldsymbol{Z}\boldsymbol{Z}^\top\right)}_{\text{Expansion: } \mathcal{R}(\boldsymbol{Z}, \epsilon)}$$

$$- \underbrace{\sum_{i=1}^K \frac{\text{tr}(\boldsymbol{\Pi}_i)}{2m} \log\det\left(\boldsymbol{I} + \frac{d}{\text{tr}(\boldsymbol{\Pi}_i)\epsilon^2} \boldsymbol{Z}\boldsymbol{\Pi}_i\boldsymbol{Z}^\top\right)}_{\text{Compression: } \mathcal{R}_c(\boldsymbol{Z}, \boldsymbol{\Pi}, \epsilon)}. \tag{1}$$

In this formulation, the membership matrices $\boldsymbol{\Pi}$ may either be fixed by labels (supervised case) or optimised jointly with $\boldsymbol{Z}$ (unsupervised case). This flexibility enables MCR$^2$ to unify both paradigms within a single framework. The coding precision $\epsilon > 0$ is typically and heuristically chosen to be very small so that all fine structural details of data are preserved in learned features.

**Coding rate.** A central component of MCR$^2$ is the *coding rate* $\mathcal{R}(\cdot, \epsilon)$, which quantifies the effective volume of a distribution or its finite sample set under a prescribed distortion level $\epsilon > 0$ (Rissanen, 1998; Cover, 1999; Ma et al., 2007). Formally, for each $\boldsymbol{z} \in \boldsymbol{Z}$, let its reconstruction $\widehat{\boldsymbol{z}}$ satisfy

$$\mathbb{E}[\|\boldsymbol{z} - \widehat{\boldsymbol{z}}\|] \leq \epsilon,$$

the average number of binary bits required to encode the feature set $\boldsymbol{Z}$ is given by $\mathcal{R}(\boldsymbol{Z}, \epsilon) = \frac{1}{2} \log\det\left(\boldsymbol{I} + \frac{d}{m\epsilon^2} \boldsymbol{Z}\boldsymbol{Z}^\top\right)$. This expression admits a clear geometric interpretation: it represents the volume of the subspace spanned by $\boldsymbol{Z}$, measured in units of $\epsilon$-balls (i.e., $d$-dimensional spheres of radius $\epsilon$). Intuitively, a larger coding rate indicates that more $\epsilon$-balls are required to cover the feature subspace, implying a richer feature set. This closed-form formulation, originally derived for Gaussian data supported on a subspace (Ma et al., 2007), offers both computational tractability and geometric as well as statistical interpretability within the MCR$^2$ framework.

**Normalization and geometric view.** The coding rate is closely related to the *volume* spanned by the features. If the features are arbitrarily scaled, the measured volumes are no longer comparable across

classes. To ensure fairness, Yu et al. (2020) normalise the scale of each class such that $\|\boldsymbol{Z}_i\|_F^2 = m_i$, a condition that can be conveniently enforced using batch normalization during training.

From this perspective, the two terms in equation 1 provide a natural geometric interpretation. The expansion term $\mathcal{R}(\boldsymbol{Z}, \epsilon)$ measures the overall volume of the feature set $\boldsymbol{Z}$; maximizing it encourages the features to spread out and occupy as large a region of the space $\mathbb{R}^d$ as possible. The compression term $\mathcal{R}_c(\boldsymbol{Z}, \boldsymbol{\Pi}, \epsilon)$ measures the volume of features within each class; minimizing it reduces the within-class spread, pulling samples of the same label into a compact, low-dimensional cluster.

Yu et al. (2020); Chan et al. (2022) showed that optimizing the overall objective yields representations with two key properties: (i) features within each class concentrate on a linear subspace that reflects the underlying submanifold, while subspaces of different classes tend toward orthogonality, thereby enhancing discriminability; and (ii) with sufficiently high coding precision (e.g., $\epsilon^2 = 0.5$), the subspaces collectively expand to span the full dimensionality of the feature space $\mathbb{R}^d$, i.e.,

$$\text{rank}(\boldsymbol{Z}) = \sum\nolimits_{i=1}^{K} \text{rank}(\boldsymbol{Z}_i) = \sum\nolimits_{i=1}^{K} d_i = d,$$

so that each class preserves the maximal possible structural components in its feature set $\boldsymbol{Z}_i$.

However, we demonstrate that in the supervised setting, blindly maximizing structural details within $\text{MCR}^2$ can make the model overly sensitive to input noise (Table 2) and may even lead to severe underfitting and poor generalization (Figure 1 and Table 1). To mitigate these issues, we argue that it is sufficient for $\text{MCR}^2$ to capture only the most discriminative structural component of each class, i.e., $\text{rank}(\boldsymbol{Z}_i) = 1$. This view resonates with the information bottleneck theory (Tishby & Zaslavsky, 2015; Hu et al., 2024), which states that the role of a neural network is to extract features $\boldsymbol{Z}$ that retain only the minimal sufficient information relevant to the target labels while discarding irrelevant details. In a similar spirit, but from a different perspective, the $\text{MCR}^2$ framework aims to capture discriminative low-dimensional structures; thus, in the supervised case, it may be sufficient to preserve only the minimal structural information required for class separation.

To learn single-component discriminative representations via $\text{MCR}^2$, we present Theorem 1 to characterize how the coding precision $\epsilon$ influences the dimensionality of each subspace:

**Theorem 1.** *Let $\boldsymbol{Z}^* = \boldsymbol{Z}_1^* \cup \cdots \cup \boldsymbol{Z}_K^*$ be the optimal solution to equation 1. Define $d_i^\star = \sqrt{\frac{m_i}{m}} \frac{d}{\epsilon^2}$. Then the following properties hold:*

- **Discriminativeness:** *Features from different classes reside in mutually orthogonal, low-dimensional linear subspaces; that is, $(\boldsymbol{Z}_i^*)^\top \boldsymbol{Z}_j^* = \boldsymbol{0}$ for all $i \neq j$.*

- **Bounded Dimensionality:** *Each class-specific subspace has dimensionality $d_i \leqslant d_i^\star$. Furthermore, for each class, the first $d_i - 1$ singular values of $\boldsymbol{Z}_i^*$ are identical.*

This theorem implies that the dimensionality $d_i$ of each class-specific subspace cannot exceed $d_i^\star$. Recall that $\|\boldsymbol{Z}_i\|_F^2 = m_i = \sum_{j=1}^{\min(d,m_i)} \sigma_j^2$ where $\sigma_j$ denotes the j-th singular value of $\boldsymbol{Z}_i$. As $d_i^\star$ decreases, $\text{rank}(\boldsymbol{Z}_i)$ also decreases, causing the energy to concentrate on fewer singular values. This, in turn, highlights the significance of the remaining structural components. Now we can encourage each class to collapse toward its minimal one-dimensional subspace, i.e., $d_i = \text{rank}(\boldsymbol{Z}_i) \rightarrow 1$. It should be emphasized that when the data are imbalanced, a uniform coding precision $\epsilon$ cannot simultaneously enforce all subspaces to be one-dimensional, since the dimensionality of each class subspace is also influenced by its proportion in the dataset, i.e., $\frac{m_i}{m}$. Therefore, we require the weaker condition $\min_{1 \leq i \leq K} d_i \geqslant 1$. From this, an upper bound for $\epsilon$ can be readily obtained:

$$\epsilon^2 \leqslant \epsilon_U^2 = \min_{1 \leq i \leq K} d\sqrt{\frac{m_i}{m}}.$$

Theoretically, the feature set $\boldsymbol{Z}_i$ of each class can be constrained to retain only a single discriminative structural component by setting $\epsilon = \epsilon_U$. However, such a constraint may be overly restrictive in practice, especially for complex deep neural networks with nonlinear objectives, where perfect convergence is rarely attainable on challenging datasets. For instance, Zhu et al. (2021) showed that the global optimality conditions for cross-entropy with certain regularization terms require the existence of at least one redundant dimension in the representation space, along which the gradient can escape local minima. Motivated by this analysis, we advocate setting the minimum dimensionality of

each subspace to $d_i \geqslant 2$. We emphasize that although Wang et al. (2024) provided a global landscape analysis for a variant of MCR$^2$, their formulation differs from ours, and their optimality condition relies on maximising structural details, offering limited guidance for our setting. Moreover, since $d_i^\star$ may not serve as a strict upper bound except in the trivial case $d_i^\star = d_i = 1$, ensuring $d_i = 2$ requires $d_i^\star > 2$. Consequently, we recommend adopting $d_i^\star = 3$, which provides both a stronger theoretical guarantee and greater practical robustness.

Accordingly, the practically upper bound for $\epsilon$ is

$$\epsilon^2 \;\leqslant\; \epsilon_\star^2 = \min_{1 \leq i \leq K} \frac{d}{3} \sqrt{\frac{m_i}{m}}.$$

As demonstrated in our experiments in Section 3, setting $\epsilon = \epsilon_\star$ is sufficient for the model to learn single-component discriminative representations in practice. Further increasing $\epsilon$ does not provide additional benefit.

Building on above theoretical insights, we propose a paradigm shift from conventional MCR$^2$, which seeks to capture maximal structural components, toward focusing on the most discriminative component. We refer to this variant as ***Single-component Maximal Coding rate reduction*** (SiMCoding):

$$\max_{\boldsymbol{Z},\boldsymbol{\Pi}} \Delta\mathcal{R}(\boldsymbol{Z},\boldsymbol{\Pi},\epsilon) = \mathcal{R}(\boldsymbol{Z},\epsilon) - \mathcal{R}_c(\boldsymbol{Z},\boldsymbol{\Pi},\epsilon), \qquad \text{s.t.} \quad \epsilon^2 = \min_{1 \leq i \leq K} \frac{d}{3} \sqrt{\frac{m_i}{m}}. \tag{2}$$

**Positioning SiMCoding among MCR$^2$ and CRATE.** MCR$^2$ can be directly employed as a loss function to train predefined neural networks such as ResNet-18 (He et al., 2016), which is arguably the most straightforward way to utilize it. Beyond this, Chan et al. (2022) demonstrated that a deep neural network can be interpreted as the unrolling of iterative gradient steps for optimizing MCR$^2$, where each layer corresponds to one iteration. This perspective enables the principled design of *white-box neural networks*. In particular, when shift-invariance is enforced for classification, the resulting architecture naturally takes the form of a multi-channel convolutional network, termed *ReduNet*. A key limitation, however, is that constructing ReduNet is computationally demanding, which hinders its scalability. Consequently, Chan et al. (2022) primarily introduced ReduNet as a rigorous proof-of-concept, with validation limited to small-scale datasets such as MNIST (LeCun, 1998). Nonetheless, this work has been highly influential.

Building on this line of research, Yu et al. (2023; 2024a) introduced sparse MCR$^2$ and approximation techniques, giving rise to a white-box transformer-like architecture termed *CRATE*, which has since been applied across diverse domains (Pai et al., 2023; Yu et al., 2024b; Yang et al., 2024). However, CRATE requires learning class-specific sets of orthonormal bases $\boldsymbol{U}_k$, which in practice demands large-scale datasets for effective training. In CRATE, the features are projected into these low-dimensional basis spaces $\boldsymbol{U}_k$. Since $\boldsymbol{Z}$ is sparse, the class-specific projection $\boldsymbol{U}_k^\top \boldsymbol{Z}$ has lower dimensionality than $\boldsymbol{U}_k$ itself. Interestingly, in our experiments we observed that CRATE also tends to learn nearly one-dimensional, mutually orthogonal subspaces, albeit through a mechanism fundamentally different from that of SiMCoding.

**Computational Complexity.** The computational complexity of SiMCoding is dominated by the computation of $(K + 1)$ log-determinants, resulting in a total cost of $\mathcal{O}\big(K \min(d^3, m^3)\big)$. This indicates that SiMCoding is most suitable for datasets with a moderate number of classes.

It can be concluded that, in the supervised setting, MCR$^2$ remains constrained to relatively simple datasets such as MNIST, whereas CRATE represents a significant step toward scalability on large-scale datasets such as ImageNet-1K and ImageNet-21K (Deng et al., 2009). Our proposed SiMCoding strikes a balance, being particularly well-suited for datasets with a moderate number of classes (e.g., $K \leq 100$), where it achieves superior overall performance compared not only to MCR$^2$ and CRATE but also to cross-entropy, especially in the presence of label or feature noise.

## 3 EXPERIMENT

In this section, we evaluate SiMCoding in terms of fitting ability, generalization, and robustness to feature noise, label noise, and catastrophic forgetting. Our aim is not to exhaustively explore extensions or engineering refinements, but rather to demonstrate that even the simplest use of SiMCoding

Table 1: Training and test accuracy (%) of different methods across datasets. Best results are in bold.

| Method | MNIST | | CIFAR-10 | | CIFAR-20 | | CIFAR-100 | | ImageNette | |
|---|---|---|---|---|---|---|---|---|---|---|
| | Train | Test | Train | Test | Train | Test | Train | Test | Train | Test |
| CE | **100.00** | 99.08 | **99.98** | **93.23** | **99.95** | 80.76 | **99.98** | **75.14** | **97.99** | 91.78 |
| MCR$^2$ | 99.81 | 98.41 | 93.83 | 90.83 | 43.66 | 41.89 | 6.15 | 6.27 | 56.72 | 51.15 |
| CRATE | 99.70 | 96.01 | 93.88 | 79.90 | 93.01 | 53.45 | 98.36 | 51.23 | 92.17 | 82.19 |
| SiMCoding | 99.94 | **99.09** | 99.32 | 93.04 | 96.15 | **80.87** | 98.90 | 74.10 | 95.70 | **92.85** |

provides a strong alternative for moderate-scale classification tasks. We compare against influential baselines—cross-entropy (CE), MCR$^2$, and CRATE—focusing on validating the effectiveness of the SiMCoding principle under fair conditions. Additional implementation details and experiments are provided in the Appendix, with code included in the supplementary material to reproduce all results.

**Performance Metric.** Traditional MCR$^2$-based methods typically rely on a Nearest Subspace Classifier (NSC), which assigns labels by measuring the distance of a feature representation to the principal subspace of each class (Yu et al., 2020; Chan et al., 2022). For class $i$, let $\boldsymbol{\mu}_i$ be the class mean and $\boldsymbol{U}_i$ the matrix containing its top $r_i$ principal components. Given a test feature $f(\boldsymbol{x}'; \boldsymbol{\theta})$, classification is performed by finding the subspace that minimizes the projection error:

$$i' = \arg \min_i \left\| (\boldsymbol{I} - \boldsymbol{U}_i \boldsymbol{U}_i^\top)(f(\boldsymbol{x}'; \boldsymbol{\theta}) - \boldsymbol{\mu}_i) \right\|_2^2.$$

In contrast, for our SiMCoding, the dimensionality $d_i$ of each class subspace is designed to approach one, making it non-trivial to predefine a fixed $r_i$. As a result, the NSC is not applicable in this setting. Instead, we evaluate both SiMCoding and MCR$^2$ by training a simple logistic softmax classifier on their learned features and reporting its accuracy as the performance measure.

### 3.1 FITTING AND GENERALISATION

**Dataset.** In this subsection, we study the fitting and generalization ability of SiMCoding. We evaluate on MNIST (LeCun, 1998), the CIFAR family (Krizhevsky et al., 2009) including CIFAR-10, CIFAR-20 (the coarse-label version of CIFAR-100), and CIFAR-100, as well as ImageNette (Howard, 2019), a 10-class subset of ImageNet.

**Architecture and Training.** For the MNIST dataset, we use a compact convolutional network consisting of two $3 \times 3$ convolutional layers (with 32 and 64 channels, both with ReLU activation), followed by $2 \times 2$ max pooling, a fully connected ReLU layer with 1024 units, and a final projection to $d = 64$ dimensions. In all experiments, the learned features are normalized such that $\|\mathbf{Z}_i\|_F^2 = m_i$ for each class. For networks trained with CE, we append a classification layer to the same backbone architecture used for MCR$^2$ and SiMCoding. For the CIFAR datasets and ImageNette, we employ a ResNet-18 (He et al., 2016) backbone, replacing the final layer with a two-layer ReLU-activated MLP that outputs 512-dimensional representations. To ensure consistency and comparability, we adopt training hyperparameters closely following (Yu et al., 2020; Chan et al., 2022; Yu et al., 2024a). Implementation details are in the Appendix.

**Performance Comparison.** Table 1 report the final training and test accuracy, and Figure 5 in the Appendix illustrates learning dynamics across datasets. For training accuracy, CE achieves nearly perfect fitting, while SiMCoding attains a comparable level even on challenging datasets such as CIFAR-100 and ImageNette, indicating a fitting capacity on par with CE. CRATE also shows strong fitting but is weaker than CE and SiMCoding, whereas MCR$^2$ converges to much lower values, especially on CIFAR-20 and CIFAR-100. In terms of test accuracy, SiMCoding consistently ranks among the best. It matches CE on MNIST and CIFAR-10, while slightly surpassing it on CIFAR-20 and ImageNette, suggesting stronger robustness as dataset complexity increases. CRATE underperforms on test accuracy (e.g., $51.23\%$ on CIFAR-100), reflecting its reliance on class-specific orthonormal bases $\boldsymbol{U}_i$, which work better with large-scale data. MCR$^2$ yields the weakest generalization, consistent with its limited training performance. Overall, SiMCoding combines strong fitting capacity with consistently high generalization, outperforming or matching CE and clearly surpassing CRATE and MCR$^2$ in both stability and robustness.

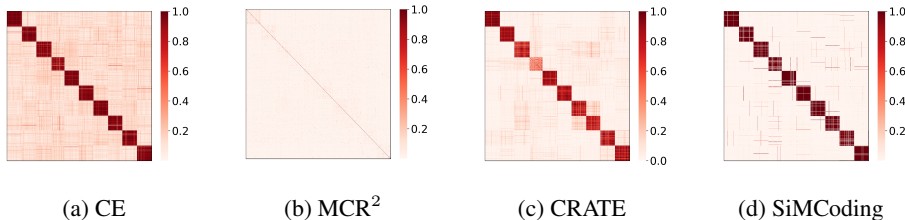

Figure 2: Heatmaps of $|\mathbf{Z}\mathbf{Z}^\top|$ on ImageNette, with samples sorted by class.

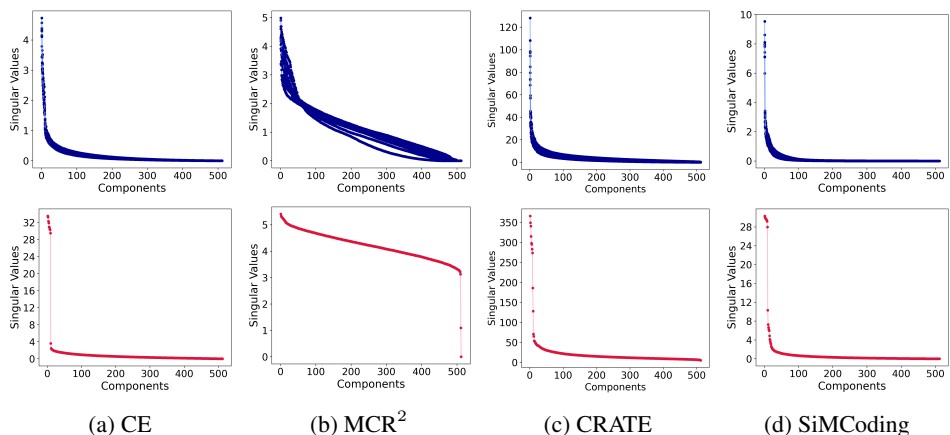

Figure 3: Feature spectra analysis on ImageNette under different training objectives. Top row: per-class singular value spectra of $\mathbf{Z}_i$. Bottom row: overall singular value spectrum of $\mathbf{Z}$.

**Structure analysis of LDR.** To examine the structure of the learned representations, we compute $\mathbf{Z}\mathbf{Z}^\top$ on ImageNette, sorting samples by class index (Figure 2). This highlights inter-class orthogonality. SiMCoding produces features that are clearly orthogonal across classes, yielding well-separated representations. CRATE also induces a block-diagonal structure, though less sharply, consistent with its slightly lower training accuracy. In contrast, MCR$^2$ fails to enforce separation, showing little block structure, while CE exhibits approximate orthogonality, consistent with neural collapse phenomenon (Papyan et al., 2020). Figure 3 shows the singular value spectra of per-class features $\mathbf{Z}_i$ and the overall matrix $\mathbf{Z}$. For CE, CRATE, and SiMCoding, each class spectrum is dominated by a single leading singular value, indicating nearly rank-one features, while MCR$^2$ retains a much flatter spectrum, suggesting high-rank intra-class variability. At the global level, CE, CRATE, and SiMCoding exhibit about ten dominant singular values, aligning with the number of classes and indicating an effectively low-rank feature space, whereas MCR$^2$ produces a full-rank spectrum, reflecting its failure to compress intra-class variation or separate classes.

Results on other datasets showing similar patterns are in Figures 7, 8, 9, and 10 in the Appendix.

**Discarding Structural details** To further investigate the learning mechanism of SiMCoding, we visualize mean saliency maps for 500 randomly selected images per class. For CE, MCR$^2$, and SiMCoding, saliency maps are computed directly. For CRATE, which relies on tokenization, we instead aggregate the four attention heads on MNIST into a single visualization. Figure 4 shows the results for digits 0 and 1, with the full set in Figure 6 in the Appendix. The differences are evident. CE produces saliency patterns that only partially align with digit structure, reflecting its focus on label fitting rather than structural abstraction. MCR$^2$ captures fine-grained details. CRATE distributes attention across digit components, capturing complementary cues. In contrast, SiMCoding concentrates on the most discriminative element—the digit outline—suggesting a mechanism that filters redundant details while preserving class-defining features.

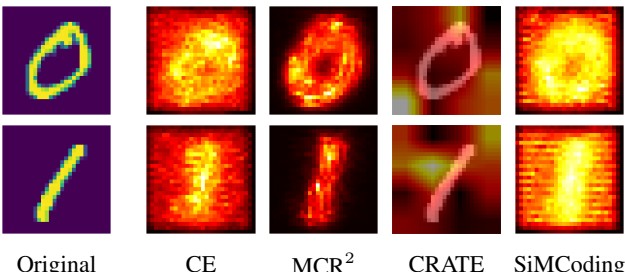

Figure 4: Comparison of saliency maps for digits 0 and 1.

Table 2: Training and test accuracy (%) on CIFAR-10 under different levels of feature noise (std).

| | Training Accuracy | | | | | Test Accuracy | | | | |
|---|---|---|---|---|---|---|---|---|---|---|
| Noise Std | 0.04 | 0.08 | 0.12 | 0.16 | 0.20 | 0.04 | 0.08 | 0.12 | 0.16 | 0.20 |
| CE | 99.91 | 99.74 | 99.38 | 98.60 | 97.57 | 91.04 | 87.94 | 84.59 | **81.04** | **78.59** |
| MCR$^2$ | 91.31 | 87.54 | 83.79 | 80.38 | 75.91 | 88.30 | 83.93 | 79.92 | 75.29 | 69.76 |
| CRATE | 92.92 | 90.48 | 87.46 | 83.48 | 79.53 | 77.64 | 71.77 | 62.66 | 54.62 | 52.81 |
| SiMCoding | 98.26 | 96.15 | 93.30 | 90.16 | 87.25 | **91.41** | **88.02** | **84.66** | 81.00 | 78.03 |

## 3.2 ROBUSTNESS AGAINST FEATURE NOISE

Following Chan et al. (2022), we corrupt CIFAR-10 with additive Gaussian noise $\mathcal{N}(0, \sigma^2)$ at $\sigma \in \{0.04, 0.08, 0.12, 0.16, 0.20\}$, keeping the architecture and training setup unchanged. Final accuracies are in Table 2, with learning dynamics in Figure 11 in the Appendix.

It is clear that test accuracy decreases monotonically with $\sigma$ for all frameworks. CE maintains nearly saturated training accuracy (e.g., $99.9\% \rightarrow 97.6\%$), reflecting its tendency to fit even heavily corrupted inputs. SiMCoding achieves the strongest or tied generalization at low–moderate noise ($\sigma \le 0.12$) and stays within $0.5\,\mathrm{pp}$ of CE at higher noise.

MCR$^2$ and CRATE degrade sharply. MCR$^2$ encodes noise along with fine details ($88.30 \rightarrow 69.76$), while CRATE is even more brittle ($77.64 \rightarrow 52.81$), reflecting its reliance on class bases $U_i$ that also capture corrupted variability. Their convergence is slower and less stable than CE and SiMCoding. Overall, CE and SiMCoding are the most robust, with SiMCoding matching or surpassing CE at moderate noise and maintaining a substantially smaller generalization gap at higher corruption levels.

## 3.3 ROBUSTNESS AGAINST LABEL NOISE

We evaluate robustness by randomly corrupting a ratio $\alpha \in \{0.1, 0.2, 0.3, 0.4, 0.5\}$ of CIFAR-20 labels. Final accuracies are reported in Table 3, with training and test dynamics in Figure 12. Training accuracy values closest to $1 - \alpha$ indicate selective fitting of correctly labeled samples.

CE attains nearly perfect training accuracy across all $\alpha$, showing that it memorizes noisy labels. Consequently, its test accuracy drops sharply ($73.6\% \rightarrow 40.6\%$ as $\alpha$ increases from 0.1 to 0.5). In contrast, SiMCoding fits mainly the correctly labeled portion, with training accuracy tracking $1 - \alpha$ (e.g., $87.9\%$ at $\alpha = 0.1$ and $49.6\%$ at 0.5). This selective fitting prevents overfitting and yields the strongest test performance, consistently surpassing CE and other baselines. CRATE shows moderate robustness in training but weak generalization ($25.4\%$ test at $\alpha = 0.5$). MCR$^2$ performs poorly overall, with training accuracy below $30\%$ even at $\alpha = 0.1$ and consistently low test results. Overall, SiMCoding demonstrates the strongest robustness to label noise, limiting fitting to reliable samples and achieving substantially better generalization than CE, CRATE, and MCR$^2$.

## 3.4 ROBUSTNESS AGAINST CATASTROPHIC FORGETTING

Chan et al. (2022) show that optimizing MCR$^2$ via iterative gradient ascent naturally induces a multi-layer white-box network, ReduNet. Wu et al. (2021) further adapt ReduNet to incremental learning,

Table 3: Training and test accuracy (%) on CIFAR-20 under different label noise ratios.

| Ratio $\alpha$ | Training Accuracy | | | | | Test Accuracy | | | | |
|---|---|---|---|---|---|---|---|---|---|---|
| | 0.1 | 0.2 | 0.3 | 0.4 | 0.5 | 0.1 | 0.2 | 0.3 | 0.4 | 0.5 |
| CE | 99.74 | 99.67 | 99.76 | 99.79 | 99.76 | 73.56 | 65.94 | 58.37 | 47.89 | 40.57 |
| MCR$^2$ | 30.03 | 23.63 | 19.71 | 15.62 | 13.76 | 30.37 | 25.06 | 22.12 | 16.37 | 14.52 |
| CRATE | 91.97 | 90.23 | 88.94 | 85.90 | 84.51 | 48.00 | 42.56 | 35.71 | 31.21 | 25.42 |
| SiMCoding | **87.86** | **80.15** | **71.53** | **61.25** | **49.63** | **77.70** | **71.26** | **64.80** | **57.83** | **50.70** |

Table 4: Test accuracy (%) on Task 1 after each training session on MNIST and CIFAR-10.

| Algorithm | MNIST | | | | | CIFAR-10 | | | | |
|---|---|---|---|---|---|---|---|---|---|---|
| | Task 1 | Task 2 | Task 3 | Task 4 | Task 5 | Task 1 | Task 2 | Task 3 | Task 4 | Task 5 |
| ReduNet | **99.95** | 98.02 | 95.82 | 94.94 | **92.95** | 78.75 | 62.35 | **48.80** | **44.50** | **43.07** |
| SiM-ReduNet | 99.91 | **98.05** | **95.94** | **94.96** | 92.25 | **83.15** | **63.45** | 47.97 | 44.11 | 40.99 |

showing reduced catastrophic forgetting. The key difference between MCR$^2$ and our SiMCoding lies in subspace dimensionality: MCR$^2$ favors maximal dimensions, whereas SiMCoding seeks minimal ones, ideally one-dimensional. This property allows SiMCoding to also yield a ReduNet-like network. We therefore propose SiM-ReduNet, obtained by optimizing SiMCoding through iterative gradient ascent, and show that it offers stronger robustness to catastrophic forgetting and improved performance over ReduNet. A limitation of ReduNet-based incremental learning is memory cost. We adopt a simplified architecture compared to Wu et al. (2021), but still demonstrate that SiMCoding can be directly applied to incremental learning while retaining robustness.

**Experimental setup.** We evaluate on MNIST and CIFAR-10 under a class-incremental setting, splitting the 10 classes into 5 tasks of 2 classes each. After each task, performance is measured on all classes seen so far. For MNIST, we follow Wu et al. (2021) but reduce the network depth to 50 layers. For CIFAR-10, we omit Gaussian kernel lifting, use a shallower 10-layer network instead of 50, and increase the learning rate from $\eta = 0.5$ to $\eta = 2.5$.

Table 4 reports test accuracy on the first task after each incremental training session. On MNIST, ReduNet and SiM-ReduNet perform similarly, both maintaining high accuracy; SiM-ReduNet is slightly stronger on intermediate tasks but marginally weaker at the final task. On CIFAR-10, forgetting is more pronounced. SiM-ReduNet starts from a higher baseline (e.g., 83.15% vs. 78.75% after Task 1) and holds an advantage in early stages, though ReduNet retains slightly more by the last task. These results confirm that SiMCoding can be effectively extended to incremental learning: SiM-ReduNet closely matches ReduNet's robustness to catastrophic forgetting and delivers stronger performance on earlier tasks.

## 4 CONCLUSION

We introduced SiMCoding, a framework that learns representations where each class is characterized by a single, most discriminative component, in contrast to the maximal structural details favoured by MCR$^2$. This shift yields strong empirical benefits: SiMCoding consistently matches or surpasses cross-entropy in generalization while showing markedly stronger robustness to label and feature noise. Compared to CRATE and MCR$^2$, it demonstrates superior fitting capacity, stability, and inter-class separation across moderate-scale benchmarks. When adapted to incremental learning, SiMCoding also shows robustness against catastrophic forgetting. Despite its computational complexity, these results establish SiMCoding as a simple yet powerful alternative to cross-entropy and related coding-based methods on moderate datasets, with strong potential for robust representation learning and incremental learning.

Empirically, we observed that both CRATE and SiMCoding pursue minimal-dimensional, mutually orthogonal subspaces. Establishing a rigorous connection between the two would be valuable for both theoretical understanding and practical application, which we leave for future work.

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
