# OpenReview forum: "Learning Single-Component Discriminative Representations via Maximal Coding Rate Reduction"
_ICLR.cc/2026/Conference — ICLR 2026 Conference Withdrawn Submission_

### Official Review · Reviewer_42QB · 2025-10-25

**Soundness:** 2
**Presentation:** 2
**Contribution:** 1
**Rating:** 2
**Confidence:** 3

**Summary:**

This paper builds on the maximal coding rate reduction ($\text{MCR}^2$) framework, which aims to capture class structures in subspaces. The authors argue that $\text{MCR}^2$’s objective can increase model sensitivity to feature noise and harm generalization. To address this, they propose retaining only the most discriminative structural component per class. The resulting approach, **SiMCoding**, is evaluated in supervised learning, white-box architectures, and incremental learning settings across multiple datasets.

**Strengths:**

1. The paper is well-structured and professionally presented, with clear organization and proper formatting.

2. The background on maximal coding rate reduction ($\text{MCR}^2$) is well-explained, helping readers follow the main idea.

**Weaknesses:**

**Major Concerns**
1. **Lack of theoretical justification**: The proposal to retain only a single discriminative component per class is not theoretically grounded. It remains unclear why or under what conditions this choice would be optimal.

2. **Conceptual inconsistency**: The narrative is at times contradictory. The authors first propose to constrain the subspace dimension to $d_i=1$, but later advocate $d_i \geq 2$ and make targeted adjustments to upper bound $\epsilon$, which undermines the core thesis.

3. **Limited novelty**: The contributions appear incremental and trivial. The main technical content seems to boil down to Eq.(2), which is $\text{MCR}^2$ with a particular choice of $\epsilon$. This modification does not convincingly offer new theoretical or practical insights beyond standard hyperparameter tuning.

4. **Weak empirical evidence**: The advantage over $\text{MCR}^2$ seems primarily due to delicately chosen $\epsilon$, while $\text{MCR}^2$ results are not tuned comparably. It’s possible that $\text{MCR}^2$ is better for some $\epsilon$. Furthermore, SiMCoding performs worse in training accuracy in Table 3, yet the paper highlights it as superior. The gains over simple cross-entropy loss in Tables 1,2,4 are negligible and not statistically persuasive, raising questions about the method’s practical value.

**Minor Issues**

1. Typo in line 95: “detailsza” → “details”.

2. The paper does not discuss its limitations.

**Questions:**

1. Why compare with CRATE in Figure 1 and Table 1? CRATE is a Transformer-like model, whereas Cross-Entropy (CE), $\text{MCR}^2$, and your SiMCoding are training objectives. How can they even be fairly comparable?

2. The superiority over $\text{MCR}^2$ is not properly explained and somewhat counterintuitive, because I feel like SiMCoding is analogous to taking only one principal component in PCA, while $\text{MCR}^2$ takes the top $d_i$ components. In principle, the latter preserves more information. Why would the reduced representation be better?

3. How do you define “feature noise”? How is that different from input noise and related to theoretical claims?

4. In line 200, the authors write "As ${d_i}^{\*}$ decreases, rank($Z_i$) also decreases, causing the energy to concentrate on fewer singular values". What is the freedom of choice to make ${d_i}^{\*}$ decrease, and why will that make rank($Z_i$) decrease? Can you explain it in detail?

5. Can you clarify more on the difference from the previous theoretical analysis on $\text{MCR}^2$ [1], and why your results are significant?

6. Have you tested performance when varying the number of structural components from 1 to $d_i$? This could empirically verify whether the argument to learn one component is sound or not. Also, could you show comparisons without corrupted labels to test whether SiMCoding can still perform well and $\text{MCR}^2$ cannot.

7. The authors’ core claim is to constrain $d_i=1$; yet, in line 126, you also advocate for $d_i \geq 2$. What messages do authors want to send eventually given this contradictory argument?

[1] A global geometric analysis of maximal coding rate reduction, ICML, 2024

---

### Official Review · Reviewer_Y4qP · 2025-10-29

**Soundness:** 2
**Presentation:** 1
**Contribution:** 2
**Rating:** 2
**Confidence:** 4

**Summary:**

The paper proposes SiMCoding (Single-component Maximal Coding Rate Reduction), a simplified version of the Maximal Coding Rate Reduction (MCR$^2$) framework.
Instead of maximizing the structural richness of each class subspace, SiMCoding enforces each class to be represented by only one dominant structural component (rank = 1).
The authors argue that this approach improves generalization and robustness to noise, while retaining the interpretability of MCR².
Experiments are conducted on small and medium-scale datasets (MNIST, CIFAR-10/20/100, ImageNette).

**Strengths:**

- The motivation is intuitive and connects to existing theory in information-theoretic representation learning.

- The paper is generally well-written and the theoretical exposition is readable

- The formulation is simple to implement and can serve as a baseline or diagnostic tool to study over-structured representations in MCR²-style methods.

**Weaknesses:**

- **limited novelty** - The proposed method is essentially a constrained / degenerate case of the MCR² objective (rank = 1 per class).
No new principle, optimization scheme, or theoretical insight is introduced beyond existing MCR² work

- **Inconsistent evaluation** - For instance, in Table 3, although SiMCoding does not always achieve the highest scores, all reported metric values are bounded, which seems methodologically inconsistent. If the goal is to demonstrate generalization ability or control of overfitting, reporting the performance gap Δ(train – test) would be more appropriate. Moreover, since small datasets are often sensitive to hyperparameters and random seeds, it would be helpful to include analyses such as performance-versus-epoch curves or statistics over multiple runs.

- **marginal benefit** -
Overall accuracy improvements over CE are marginal (< 0.3%) in most datasets.
The method performs worse than ReduNet when embedded as SiM-ReduNet (see Table 5), contradicting the claim that the simplification improves learning robustness.

**Questions:**

- On SiM-ReduNet: Table5 shows SiM-ReduNet performing worse than the original ReduNet. Does this suggest that enforcing rank-1 subspaces restricts representational capacity? Have the authors examined intermediate feature representations to verify this effect?
- Overfitting measurement: SiMCoding is claimed to reduce overfitting, yet in \textbf{Table~2}, the original MCR$^2$ shows a smaller train–test gap ($\Delta$) than SiMCoding. Could the authors clarify this discrepancy and explain how overfitting is quantified?

- Applicability to large-scale data: The paper mentions computational constraints when $K > 100$. Could the authors elaborate on how SiMCoding might be approximated or accelerated for larger-scale datasets (e.g., ImageNet)?

---

### Official Review · Reviewer_72ZJ · 2025-11-01

**Soundness:** 2
**Presentation:** 3
**Contribution:** 2
**Rating:** 4
**Confidence:** 3

**Summary:**

The paper argues the viewpoint that Maximal Coding Rate Reduction (MCR2) offers a promising theoretical framework for interpreting modern deep networks through the lens of data compression and discriminative representation. Then it formulates SiMCoding framework, which constrains each class representation to a single discriminative component by fixing the coding precision.
The paper is conceptually clear and extends a known framework in an interesting direction, but the novelty and empirical evidence are limited.

**Strengths:**

1. The idea of constraining each class to a single discriminative direction is simple and interesting, and it offers a fresh perspective on supervised feature compression.
2. The paper provides a clear conceptual connection between MCR² and the information bottleneck principle.
3. Experiments cover several datasets (MNIST, CIFAR-10/20/100, ImageNette) and analyze performance under noise, providing some robustness insights.

**Weaknesses:**

1. The method is a minor specialization of the previous MCR² framework, primarily by fixing $\epsilon$ to enforce a low-rank constraint. There are no new optimization schemes, architectures, or objective formulations beyond this adjustment.
2. The experimental study uses only a few small-to-medium datasets and simple models (e.g., ResNet-18). The lack of larger-scale experiments or diverse tasks limits the generality of the conclusions.

**Questions:**

1. Sensitivity of $\epsilon$: Since $\epsilon$ is important to characterize the novelty of this work, how exactly is the value of $\epsilon$ selected in practice? Is it fixed analytically as suggested by Theorem 1, or tuned empirically? Please provide quantitative evidence showing how sensitive model performance is to different  $\epsilon$  values.
2. The current experiments are limited to small and medium datasets.Have you tested the method on larger or more complex benchmarks (e.g., ImageNet-1K or long-tailed datasets)?  If not, what practical or computational barriers prevent such scaling?
3. (Minor )How would SiMCoding behave on imbalanced datasets?

---

### Official Review · Reviewer_Dkyq · 2025-11-01

**Soundness:** 2
**Presentation:** 3
**Contribution:** 3
**Rating:** 6
**Confidence:** 3

**Summary:**

This work proposes an improvement to MCR^2, a framework for training and interpreting deep networks based on a pair of compression and expansion objectives. These objectives try to features from different classes on mutually incoherent subspaces while maximizing expressivity within each subspace. The authors show how this second objective of MCR^2 hurts generalization and makes models highly vulnerable to noisy data/labels. They then propose a fix, "SiMCoding", which fixes the MCR^2 precision parameter to enforce rank-1 solutions for each class.

Their solution is verified empirically on a wide range of discriminative tasks: MNIST, CIFAR-10, CIFAR-20, CIFAR-100, and ImageNette.

**Strengths:**

I found this paper very well-written. The idea is simple, and the results are convincing. The authors clearly motivate the problem that over-encoding intra-class variation in MCR^2 can hurt generalization; the proposed “single-component” fix is intuitively appealing.

The experimental section is thorough across several datasets and convincingly shows improved robustness to label and feature noise. The paper also benefits from good organization, clear figures, and an intuitive discussion linking geometric compression to generalization.

Overall, I felt that the work provides a useful perspective on how controlling coding precision (and feature diversity more generally) affects representation dimensionality, while offering a practical variant of MCR^2 that performs competitively with cross-entropy.

**Weaknesses:**

Most weaknesses of the method are inherited from standard MCR^2. It seems difficult to scale for computational reasons due to the log-determinant computation (hence only experiments on smaller image datasets); it's also unclear to me in real-world settings if orthogonality in feature space is the "correct" objective, particularly when multiple classes share similarities (eg, should 'bicycle' and 'motorcycle' be orthogonal).

The theoretical justification seems slightly weaker than my initial impression from the paper's introduction. Theorem 1 is framed as a rigorous relationship between $\epsilon$ and subspace dimensionality, but the actual choice of precision is in part justified heuristically in the appendix. The paper frames this as "theory removes the need to tune $\epsilon$" (lines 100-102), but the final solution still seems to have been reached empirically: setting $d_i = 3$ allows subspaces up to 3 dimensions, despite the goal being 1 dimension. The appendix acknowledges this gap between theory and practice, but it wasn't clear to me from the main paper. Assuming I'm understanding correctly, it would be nice to clarify!

**Questions:**

How sensitive is performance to the factor of 3 in $\epsilon$? Have you tried other values (1, 2, 4, 5)?

---

### Note · Authors · 2025-11-20

I have read and agree with the venue's withdrawal policy on behalf of myself and my co-authors.